# An Update on Connexin Gap Junction and Hemichannels in Diabetic Retinopathy

**DOI:** 10.3390/ijms22063194

**Published:** 2021-03-21

**Authors:** Jorge González-Casanova, Oliver Schmachtenberg, Agustín D. Martínez, Helmuth A. Sanchez, Paloma A. Harcha, Diana Rojas-Gomez

**Affiliations:** 1Instituto de Ciencias Biomédicas, Facultad de Ciencias de la Salud, Universidad Autónoma de Chile, Santiago 8910060, Chile; jorge.gonzalez@uautonoma.cl; 2Centro Interdisciplinario de Neurociencia de Valparaíso, Instituto de Biología, Facultad de Ciencias, Universidad de Valparaíso, Valparaíso 2360102, Chile; oliver.schmachtenberg@uv.cl; 3Centro Interdisciplinario de Neurociencia de Valparaíso, Instituto de Neurociencia, Facultad de Ciencias, Universidad de Valparaíso, Valparaíso 2360102, Chile; agustin.martinez@uv.cl (A.D.M.); helmuth.sanchez@cinv.cl (H.A.S.); paloma.harcha@cinv.cl (P.A.H.); 4Escuela de Nutrición y Dietética, Facultad de Medicina, Universidad Andres Bello, Santiago 8370146, Chile

**Keywords:** diabetic retinopathy, connexin, gap junction channels, hemichannels

## Abstract

Diabetic retinopathy (DR) is one of the main causes of vision loss in the working age population. It is characterized by a progressive deterioration of the retinal microvasculature, caused by long-term metabolic alterations inherent to diabetes, leading to a progressive loss of retinal integrity and function. The mammalian retina presents an orderly layered structure that executes initial but complex visual processing and analysis. Gap junction channels (GJC) forming electrical synapses are present in each retinal layer and contribute to the communication between different cell types. In addition, connexin hemichannels (HCs) have emerged as relevant players that influence diverse physiological and pathological processes in the retina. This article highlights the impact of diabetic conditions on GJC and HCs physiology and their involvement in DR pathogenesis. Microvascular damage and concomitant loss of endothelial cells and pericytes are related to alterations in gap junction intercellular communication (GJIC) and decreased connexin 43 (Cx43) expression. On the other hand, it has been shown that the expression and activity of HCs are upregulated in DR, becoming a key element in the establishment of proinflammatory conditions that emerge during hyperglycemia. Hence, novel connexin HCs blockers or drugs to enhance GJIC are promising tools for the development of pharmacological interventions for diabetic retinopathy, and initial in vitro and in vivo studies have shown favorable results in this regard.

## 1. Introduction

Worldwide diabetes prevalence is rising steadily, affecting currently close to 500 million adults. One of the most debilitating complications of diabetes is diabetic retinopathy (DR), which affects about one third of Type 1 and Type 2 Diabetes Mellitus patients, especially the population aged ≥ 40 years [1,2].

In uncontrolled diabetes, DR often proceeds to near-complete blindness through macular edema, vitreal hemorrhage or retinal detachment, and even under strict blood glucose and ophthalmologic control, it leads to severe visual impairment [3,4]. DR has traditionally been diagnosed and staged according to eye fundus evaluation, which reveals characteristic alterations of the retinal blood supply [5]. These allow a differentiation into two sequential clinical stages: First (1), a non-proliferative stage characterized by hypertrophy, abnormal permeability of the blood-retinal barrier (BRB) due to loss of integrity in the retinal microvascular endothelium, accompanied by edema, closure of capillaries, decreased perfusion, ischemia, and inflammation. This stage can be asymptomatic or associated with mild to moderate visual impairment. Then (2), due to the decrease of blood flow and nutrients, DR progresses to a proliferative stage characterized by the formation of new pathological blood vessels. However, neovascularization can also lead to hemorrhage and scar tissue formation on the surface of the retina [6]. Since the scar tissue also attaches to the vitreous gel, it can eventually pull the retinal off (tractional retinal detachment), causing together with hemorrhage severe or complete blindness [7].

Three types of early visual impairment have been described in diabetics and are attributed to initial stages of DR: Diminished dark adaptation, reduced contrast sensitivity, and color vision deficits [8,9,10,11,12]. Dark adaptation and sensitivity curves are frequently shifted in diabetics with early-stage DR, reflecting impaired night vision [13]. It is commonly assumed that rod photoreceptors, which are among the cells with the highest energy and oxygen consumption of the body, are the first affected neurons in the retina [14], due to a lack of adequate oxygen supply or increased free radical exposure [15,16]. Since the human retina, like the rodent retina, is highly rod dominated, the death of a minor percentage of rods may at first go unnoticed, but show up on sensitive electrophysiological tests, for example in the scotopic electroretinogram (ERG) b-wave [17,18]. Another symptom of early alterations in the diabetic retina is reduced blue light sensitivity and yellow–blue (tritan) contrast discrimination, which is thought to be caused by selective S-cone degeneration [19,20,21]. However, the most conspicuous alterations in early DR are found in the oscillatory potentials of the ERG and may be observed in diabetics even before a fundus-based DR diagnosis can be made [22]. These findings support the notion that diabetes-elicited changes in the inner retina, notably of bipolar and amacrine cells, the sources of the ERG b-wave and oscillatory potentials, may be involved in dark adaptation and night vision deficits. Along these lines of evidence, α-amino-3-hydroxy-5-methyl-4-isoxazolepropionic acid (AMPA) receptors, a subgroup of ionotropic glutamate receptors that regulate fast excitatory transmission by gating the flow of Ca^+2^ and Na^+^ into the cell, are altered on AII and A17 amacrine cells in early streptozotocin-induced type I diabetes in rats, which might contribute to decreased scotopic sensitivity [23,24]. The aforementioned studies allowed to conclude that early DR neuroretinal alterations are not limited to photoreceptors, but can also involve alterations in bipolar and amacrine cells of the inner retina.

Accordingly, many histological investigations in animal models and humans have found a decrease in the width of retinal layers, excluding swelling caused by edema [25,26], and a general increase in terminal deoxynucleotidyl transferase dUTP nick end labeling (TUNEL) positive cells [27,28], suggesting sporadic apoptotic cell death in both the outer and inner retina under DR conditions.

Concomitantly, there is alteration in autophagy mechanisms and differential changes in the dendritic arborization of ganglion cells and induction of apoptosis, which contributes to the reduction of the inner retinal layer [29,30]. The mechanism of action of DR on the inner layers of the retina is controversial, and there is debate about whether it is intrinsic in nature of the neural network or is the consequence of a failure in local endothelial function [31]. Key paracrine signaling events have been reported, including alteration in trophic signaling by brain derived neurotrophic factor (BDNF), nerve growth factor (NGF) and other molecules associated with signaling cascades, such as p38 mitogen-activated protein kinases (p38-MAPK), Protein kinase B (Akt), and intrinsic generation of free radicals which contribute to loss of function and cell death of ganglion cells [32,33,34,35].

## 2. Cell Biology and Pathophysiological Mechanisms in Diabetic Retinopathy

To date, the mechanisms through which neurodegeneration occurs in DR are not completely understood [3,5,36,37]. However, there is consensus that hyperglycemia is the main factor triggering the development of DR, since strict control of glycemia in diabetics prevents, at least to some degree, the development of DR [36]. Conversely, diverse animal models made hyperglycemic have been shown to develop DR-like symptoms [37,38]. Retinal oxygen and glucose consumption are among the highest in the body. Glucose reaches the retina from blood vessels in the choroid and the inner retina, in which endothelial cells express the glucose transporter GLUT1 [39]. The transporters GLUT1 and GLUT2 are also expressed in Müller cells, while GLUT3 is found exclusively in neurons in the inner plexiform layer [40]. Several biochemical pathways linking high glucose levels with pathology have been implicated in DR in humans and animal models. Among them, of special importance are excessive protein glycation, leading to the production of advanced glycation end products (AGEs), glutamate excitotoxicity, neuroinflammation, and oxidative/nitrosative stress [41,42]. Glucose reacts spontaneously with protein amino groups in the Maillard reaction, and the products of this reaction may undergo subsequent oxidation and crosslinking reactions, leading to the formation of AGEs [43]. Although AGEs have been shown to be involved in DR, presumably through BRB disruption [44,45], to the best of our knowledge specific effects of AGEs on retinal neurons have not been described. In addition, advanced lipoxidation end products (ALEs), play a role in Müller glial dysregulation, correlating with inflammation, oxidative stress, and glial cell activation [46,47,48,49].

An emerging molecular player in the pathological processes leading to DR are connexins, organized either as gap junction channels (GJCs) or hemichannels (HCs) (Figure 1). Connexins (Cxs) are present in the five neuronal types of the retina, including photoreceptors, horizontal cells, bipolar, amacrine and ganglion cells, and in glial cells, such as Müller cells and astrocytes (Table 1, Figure 1). In addition, in the retinal vascular tissue several connexins are expressed in endothelial cells and pericytes. Although immunohistological and functional studies in mice and humans have demonstrated the presence of several connexins in retinal cells (Table 1, Figure 1) [50,51,52,53], the participation of neuronal retinal connexins in DR have not been directly evaluated. Therefore, in this review, we will focus mainly on the role of connexins in vascular and glial cell in DR.

## 3. Structure, Function, and Regulation of GJC and HCs

GJCs are intercellular channels that allow the direct exchange of small molecules and ions, such as amino acids, nucleotides (ATP, GTP, ADP), second messengers (Ca^2+^, cAMP, cGMP, IP3), and various metabolites (NAD^+^, NADPH) between the cytoplasm of adjacent cells. GJCs are formed by the apposition of two hemichannels (HCs) (connexons), positioned in each of the appositional cell membranes of two contacting cells [91] (Figure 2). In addition, HCs located in non-appositional plasma membrane can constitute a way of paracrine communication and molecule exchange between the extracellular medium and the cytoplasm [92,93]. The intercellular communication mediated by GJCs is essential for the maintenance of cellular homeostasis, but they also participate in embryogenesis, cell differentiation, hormonal and paracrine signaling, neuronal electrochemical responses, cardiac physiology, and sensory physiology, in particular hearing and vision. Therefore, mutations or alterations in connexin expression and function are related to different human pathologies, like cardiovascular disease, cancer, deafness and diabetes, among others [94,95].

At least 21 connexin genes have been identified in humans so far, and 20 connexin genes have been reported in mice. Connexins are expressed in almost all tissues, except in erythrocytes, mature sperm cells and specialized skeletal muscular cells [96,97]. GJCs can be homomeric/homotypic, if they are conformed by the same connexin subunits in both appossitional HCs, and homomeric/heterotypic, if each HC in a GJC contains different connexins [98,99,100,101,102] (Figure 2). Cells commonly express more than one connexin isoform, allowing the formation of heteromeric HCs. However, although there is a high possibility of forming heteromeric channels, not all isoforms are compatible with each other; for example, in vascular cells Cx40 is compatible with Cx43, forming heteromeric GJC [103]. Furthermore, the possibility of connexins forming various types of channels allows for greater diversity in their function and regulation, by creating channels with specific gating and permeability properties. The aforementioned Cx40 and Cx43 heteromeric channel is more sensitive to pH [96]. Additionally, heteromeric channels formed by Cx32 and Cx26 have altered permeability to cyclic nucleotides, in comparison with their respective homomeric counterparts [104].

Connexin phosphorylation also plays a key role in the regulation of gap junctions and HCs function, altering the charge, hydrophobicity, and interaction with other proteins, which can influence both channel trafficking and activity. The phosphorylation of connexins occurs through serine/threonine kinases or tyrosine kinases, such as MAPK, protein Kinase C (PKC), protein Kinase A PKA, casein kinase or calcium/calmodulin-dependent protein kinase II (CaMKII), among others [105,106,107,108].

Moreover, HCs are non-selective plasma membrane channels that should be tightly regulated under physiological conditions, to minimize their activity and maintain plasma membrane electrochemical gradients and cellular homeostasis [109]. Normally, HCs are kept closed by extracellular Ca^2+^ and by the resting membrane potential [110,111]. However, there is growing evidence that these channels can be activated by different physiological or pathological conditions, allowing the transfer of small molecules such as Ca^2+^, ATP, glutamate, E2 prostaglandin, and NAD^+^ between intracellular and extracellular compartments [93,112,113,114,115,116,117]. Opening of HCs is modified by connexin phosphorylation or by intracellular Ca^2+^ concentrations [110,115,116]. In fact, moderate increments in intracellular Ca^2+^ can promote HC opening by direct or indirect mechanisms, such as direct Ca^2+^/Calmodulin binding to Cx43 or by the activation of protein kinases [118]. More often, HCs are activated under pathological conditions, including oxidative stress, mechanical stretch, inflammatory processes, and lower pH. Exacerbated HCs activity during pathological states can increase cell damage [117,119,120,121,122]. For instance, excessive ATP or glutamate release through HCs may promote toxicity to neighboring cells and propagate damage to distant cells, amplifying tissue deterioration during pathological conditions [121].

## 4. Vascular Connexins and GJCs under Diabetic Conditions

Retinal capillary endothelial cells and pericytes are functionally connected through widespread GJCs (Figure 1). However, in hyperglycemic conditions, this communication is reportedly diminished [123]. Sato et al. [63] investigated the impact of a high glucose environment on the expression of Cx37, Cx40 and Cx43 in rat endothelial cell culture. The results showed a reduction in the expression of Cx43 and the formation of gap junctions, without an effect on Cx37 and Cx40 expression. Along these lines of evidence, Fernandes et al., [59] found that an environment with high glucose levels causes downregulation of Cx43 expression in bovine retinal endothelial cells accompanied by an increase in the phosphorylated form of Cx43, which seems to promote proteasome-dependent Cx43 degradation.

Pericytes are embedded in the walls of microvessels and have stem cell potential [124,125,126]. Human and bovine retinal pericytes show a reduced expression of Cx43 and a reduction in GJIC when cultivated under high glucose conditions. In DR, the loss of Cx43 expression is associated with the death of pericytes and the formation of acellular capillaries, which are mainly composed of extracellular matrix components and a reduced number of cells [127]. Later on, Li and Roy [128] also showed that reduced expression of Cx43 in microvascular endothelial cells causes apoptosis. Similar studies were carried out in retinal samples of diabetic patients, showing that the capillary networks presented significantly higher pericyte loss and acellular capillaries compared to non-diabetic patients. This vascular deterioration was associated with a reduced expression of Cx43, and a reduction in gap junction plaque density [129]. These studies support the hypothesis that the reduced Cx43 expression observed in DR may be central to disease progression. To address this issue, Tien et al., [130] used intravitreal injection of siRNA versus scrambled siRNA as control to reduce the expression of Cx43 in the retina. The results of these experiments show that downregulation of Cx43 with siRNA produced similar results as observed in DR, notably increased apoptosis of retinal vascular cells, formation of acellular capillaries, pericyte loss and increase in vascular leakage, suggesting that the decrease in Cx43 observed in the retina of diabetic animals is a factor involved in the alteration of the BRB [130].

Blood retinal barrier stability is largely conditioned by tight junction integrity (protein zonula occludens 1 (ZO-1) and occludin). Additionally, tight junctions can interact with Cx43, and in this manner participate in GJC aggregate organization and stability [131,132,133,134,135,136]. Nevertheless, under diabetic conditions, it is possible to observe decreased protein expression, resulting in BRB breakdown [137,138,139]. In this context, some authors have verified alterations in these structural proteins and Cx43 under DR conditions [64,140]. Tien et al. [64] used cultures of rat retinal endothelial cells exposed to high glucose or cultures transfected with Cx43 siRNA to evaluate the effect of diabetic conditions on the interaction of Cx43 with ZO-1. Their findings showed that the reduction in Cx43 expression and GJCs observed was accompanied by a significant decrease in the expression of ZO-1 and occludin and an increase in cell monolayer permeability. Interestingly, when cells were exposed to high glucose and transfected with Cx43 plasmids to increase connexin expression, monolayer permeability decreased. These results suggest a decrease in Cx43 expression causes a reduction in tight junction protein expression, resulting in increased vascular permeability.

In recent years, the presence of connexins in the mitochondrial membrane has been described in different cell types, including retinal vascular cells, hepatocytes, astrocytes, stem cells, cardiomyocytes, and brown adipose tissue [141,142,143,144,145,146]. Furthermore, alteration in mitochondrial connexins expression is related to a series of pathologies [147,148]. Various studies have supported a role of mitochondrial Cx43 as a critical modulator of apoptosis. Trudeau et al. and Carette et al. [141,149] showed in rat retinal endothelial cells cultivated under high glucose conditions that mitochondrial Cx43 expression was diminished, along with a reduction in gap junctions and accompanied by an increase in mitochondrial cytochrome c release. Proteins such as matrix metalloproteinase 2 (MMP2) have been linked to a pro-apoptotic role in capillary cells in DR, and MMP2 is activated in high glucose conditions [150]. Interestingly, experiments in bovine retinal endothelial cells exposed to high glucose concentrations and in the retina of diabetic mice have confirmed activation of MMP2, producing mitochondrial damage and decreasing mitochondrial Cx43 expression [151].

## 5. DR Affects Connexins in Retinal Glial Cells

Alterations in connexin expression and function related to DR have also been observed in retinal glial cells and could be part of general transcellular processes involved in the pathology’s progression, especially during its non-proliferative stage. Ly et al. [152] investigated the changes occurring in astrocytes, Müller cells, and retinal ganglion cells in streptozotocin-induced diabetic Sprague Dawley rats. Their experiments showed that in the early stages of diabetes, Cx43 and Cx26 are downregulated in astrocytes before cell death rates become elevated in DR. In this type of cells, Cx43 is expressed abundantly and plays a pathological role during the development of retinopathy [153]. Furthermore, a significant increase in hypoxia-inducible factor 1 (HIF1α) protein expression was observed in the ganglion cell layer, which might indicate a decrease in oxygenation [152]. Six weeks after inducing diabetes, Müller cell gliosis was observed with a concomitant reduction in neuronal function and rod photoreceptor responses. In line with the aforementioned, astrocyte apoptosis or change in cell migration and proliferation in DR was also shown to be a key factor in vascular damage, eliciting NF-κB and oxidative stress activation [154,155,156,157,158]. These results suggest that a loss of astrocytes together with a decrease in the expression of Cx26 and Cx43 are early events in the development of hypoxic conditions and eventual ganglion cell alterations in DR [152]. Furthermore, it was shown that the astrocytes from rat retina undergo a switch at the gap junction plaque of connexin composition associated with aging [159]. In old rats, astroglial Cx30 expression patterns were altered compared to young rats, with increased Cx30 protein levels. An increase of the heterogeneous gap junction plaques was observed in old animals, containing channels formed by Cx26/Cx45. These findings suggest that switching connexin type expression in astrocytes is involved in age-related physiological changes detected in neuronal cells and the retinal vasculature.

Pathophysiological conditions of neovascularization in human DR can be reproduced in mice, using a model of oxygen-induced retinopathy which consists of the exposure of P14 mice to a hypoxic environment triggering retinal pathology similar to that observed in human proliferative DR [160]. In this model, it was observed that deletion of the Cx43 gene or the blocking of GJCs has a protective effect on retinal astrocytes and limits deleterious vasculature processes in ischemic areas of the retina, leading to improved neuroretinal function [153]. Furthermore, a significant increase in the phosphorylation levels of Cx43 mediated by casein kinase was observed after hypoxia induction, which was accompanied by an increase in the number of GJCs, preceding the loss of astrocytes. The inhibition of casein kinase activity generated a protective effect against the hypoxia-induced damage in astrocytes [153]. These experiments support the idea that cellular death signals can be propagated by GJC between astrocytes increasing the retinal tissue’s damaged area. Interestingly, Cx43 HC activity was not observed, contrary to other hypoxic models [120], which could be due to the putative heteromeric composition of retinal HCs.

## 6. Role of Connexins in Inflammation during DR

DR causes a chronic inflammatory alteration in the retina, which is associated with the release of proinflammatory mediators, activation of leukocytes, and elevation of reactive oxygen species (ROS) [161,162,163,164]. These inflammatory events play a critical role in developing the early and late stages of the disease, with consequences such as the disruption of BRB, pericyte loss, and apoptosis of endothelial vascular cells mentioned earlier.

Therefore, the interest of the different research groups has focused on the mechanisms involved in the regulation of inflammatory processes and connexins in DR. Mugisho et al. [165] investigated the hypothesis that inflammation is linked with connexin expression in DR, which might worsen the pathophysiology in a high-glucose environment. They conducted experiments in primary human retinal endothelial cells and determined the expression of Cx43 in high glucose and or inflammatory conditions. Their results showed that hyperglycemic environments are not enough to alter Cx43 expression. Similarly, when cultured cells were exposed to proinflammatory cytokines, Cx43 expression was also not affected. However, under simultaneous treatment with high glucose and proinflammatory interleukins, a significant increase in Cx43 expression was observed [165].

Furthermore, using *Akita* and *Akimba* genetic mouse models for diabetes and DR, different results can be obtained. For the *Akita* diabetic mice no change in Cx43 expression in retinal ganglion cell layer and vascular endothelial cells was observed, in contrast to a two-fold increase in the *Akimba* mouse model. Akita is a natural Type-1 diabetic mouse model carrying a dominant mutation in the insulin-2 gene (*Ins2^Akita^*), which presents hyperglycemia and a diminished β cell mass [166]. On the other hand, Akimba mouse model *(Ins2^Akita^ × Vegfa^+/−^)* presents a phenotype of hyperglycemia, retinal neovascularization, and vascular endothelial growth factor (VEGF) over-expression [167]. These in vivo model results suggest that Cx43 over-expression is directly associated with the inflammatory process in DR progression, mostly observable in *Akimba* mouse [165]. Additional experiments by this group with human donor retinas demonstrated a greater Cx43 expression in DR patients’ retina, compared with age-matched patients without diabetes and DR [165]. Although these results only demonstrated the influence of high in glucose proinflammatory environments on Cx43 expression, it can be inferred that this overexpression is also related to Cx43 channel (GJC and/or HCs) function alteration under these pathological conditions.

Moreover, migration of activated microglia/macrophages has been observed in a DR rat model of DR [168] and may constitute a potential source for local proinflammatory cytokine release. These compounds, along with a high glucose environment, could increase the probability of Cx43 HCs opening.

It is widely accepted that connexins play an important role in inflammatory processes, among others through HC-mediated ATP release, which increases inflammation in different conditions, including high glucose concentration [169,170,171,172]. According to this hypothesis, it has been proposed HC activity inhibition has a protective effect against inflammation progression [43,173,174,175,176,177]. Since HCs are thought to mediate and enhance deleterious inflammatory effects, propagating signaling molecules and contributing to proinflammatory mediator release [120,178,179,180]. Different research groups are currently working on Cx–HCs modulators as potential therapeutic strategies, in addition to the currently approved treatments [181]. Specific key investigations oriented in this line are summarized in Table 2.

To test this aforementioned hypothesis, cell cultures of human adult retinal pigment epithelium cells were exposed to a Cx43-HC blocker, mimetic Peptide5 [62]. These cells were cultivated under high glucose conditions and or exposure to proinflammatory cytokines, such as IL-1 β and TNF-α. The results showed that the high glucose cultures also exposed to proinflammatory cytokines generated an increased secretion of the inflammatory mediators IL-6, sICAM-1, MCP-1, and IL-8 compared with cell cultures exposed only to proinflammatory molecules. Together, these data suggest that high glucose levels exacerbate cellular events related to inflammation. Additionally, the use of a Cx43 HC blocker prevented proinflammatory mediator production, and inhibited an increase in ATP release, suggesting that it was mediated through Cx43 HC activity.

The use of specific Cx43 HCs inhibitors as a potential therapy against retinal pigment epithelial cell barrier and outer blood-retinal integrity loss has recently been investigated [140]. In cultured human retinal pigment epithelial cells, it has been determined that application of Cx43 mimetic Peptide5 prevented a transepithelial permeability increase under high glucose and proinflammatory interleukin conditions, together with a decreased ZO-1 and Cx43 loss, normally observed in cell cultures exposed to high glucose and inflammatory conditions [140].

In DR and age-related macular degeneration, the choroidal structure’s vasculature can be weakened, an event that is associated with inflammatory processes. To evaluate the protective effect of Cx43 HC blockers, Guo’s group used a light-damaged albino rat model. It demonstrated that Peptide5 reduces the inflammatory response and has a protective effect on photoreceptor function [183]. More recently, the oral use of the benzopyran derivative Tonabersat, a gap junction modulator, has also been investigated to determine its effectiveness in protecting retinal structure and function, especially that of photoreceptors and bipolar cells in the inner retina in rat models of DR and age-related macular degeneration. The results of these experiments showed an improvement in retinal photoreceptor function and a dose-dependent decrease in classical microglial/macrophage co-expression of proinflammatory markers: Ionized calcium binding adapter molecule 1 (Iba-1), and astrocytic glial fibrillary acidic protein (GFAP) in rats treated with Tonabersat [186]. Indeed, in organotypic human retinal explants treated with high glucose together with IL-1 β and TNF-α, Tonabersat treatment prevented NLRP3 (nucleotide-binding domain leucine-rich repeat (NLR) and pyrin domain containing receptor 3) inflammasome complex aggregation (key component for cytokine production), decreased VEGF secretion, and reduced GFAP expression in both Müller cells and astrocytes [187]. Certainly, DR sign decrease after Cx43 HC inhibition in retina explants is consistent with previously reported studies. More recently, similar results were obtained by Lyon et al., in human retinal pigment epithelial cell (ARPE-19) cultures exposed to IL-1β, TNFα and high glucose. Under these experimental conditions, treatment with Tonabersat prevented IL-1β, VEGF, and IL-6 release and inhibited NLRP3 and cleaved caspase-1 complex formation [186].

Beyond pharmacological therapies focused on HCs, some authors have reported experimental results using GJC modulators as a pharmacological target. Kim et al. [190] investigated whether Cx43-mediated GJIC maintenance could prevent retinal vascular cell loss under high glucose conditions. To this end, danegaptide, a dipeptide improving conductance and GJCs coupling under stress conditions, was added to rat retinal endothelial cells culture. These experiments demonstrated that cell–cell coupling maintenance may be a useful strategy in DR to inhibit apoptosis and observed cell permeability. Possible mechanisms include stabilization of Cx43 cell membrane localization, facilitating docking between HCs to generate the GJC.

Finally, Losso et al. [191] studied trans-resveratrol effect on GJIC in retinal pigment epithelium (RPE) cells. This compound, mainly present in Pinot wines and in some fruits infected with the *Botrytis cinerea* fungus, was added to ARPE-19 cell culture in medium containing 33 mM glucose. Losso et al. observed this molecule’s dose-dependent effect on GJIC increase by inhibiting Cx43 degradation. It also inhibited VEGF, TGF-β1, cyclooxygenase-2(COX-2), IL-6, and IL-8 accumulation and PKCβ activation. Although Losso et al. did not describe trans-resveratrol’s mechanism in regulating Cx43 expression, they related this molecule to trans-resveratrol’s protective effect with the low inflammation grade under high glucose conditions.

## 7. Conclusions

Currently, DR is mostly considered as a microvascular disease with neurovascular complications, although the pathological sequence is not fully established. A series of cellular events associated with a hyperglycemic environment are involved in the pathogenesis. The functional coupling of cells by connexins plays an essential role not only in normal light processing, but also in maintaining the retina’s homeostasis. However, this coupling is altered during pathological events contributing to the spread of inflammation and retinal tissue deterioration. Experiments, both in vitro and in vivo, have shown a complex alteration of connexin expression, and GJC plaque formation in retinal astrocytes. Additionally, under hyperglycemic conditions, it is possible to find connexin remodeling processes, resulting in heterogeneous gap junction plaque increase, events that also precede cell death. Additionally, in retinal capillary endothelial cells and pericytes, connexin expression and GIJC function are significantly decreased (Figure 3). This differential and cell type dependent regulation of connexins under DR could be part of the anatomical and functional complexity of the BRB. On the other hand, Cx43 HCs play a role in the pathogenesis of chronic inflammatory diseases, and even though under normal conditions HC opening has not been detected, enhanced membrane permeability facilitated by HCs has been observed under cellular injury conditions, such as ischemic or hypoxic stress. Accordingly, a putative role of vascular Cx43 HCs in DR and visual impairment emerges, which could exacerbate retinal inflammatory processes mediated by ATP release, resulting in RPE loss of integrity. Therefore, initial pharmacologic tests with experimental connexin HC blockers aim to lessen symptoms and decelerate disease progression. It is clear that a greater understanding of the underlying mechanisms involved in the delicate regulation of connexin expression and function is needed. Moreover, a more precise comprehension of the dichotomy between GJIC reduction vs. HCs activation during DR development is required. Nevertheless, to date, in experimental models Cx43–HCs blocker therapeutic potential has been demonstrated, with improved vascular integrity of the retina and decreased inflammasome activation.

## Figures and Tables

**Figure 1 ijms-22-03194-f001:**
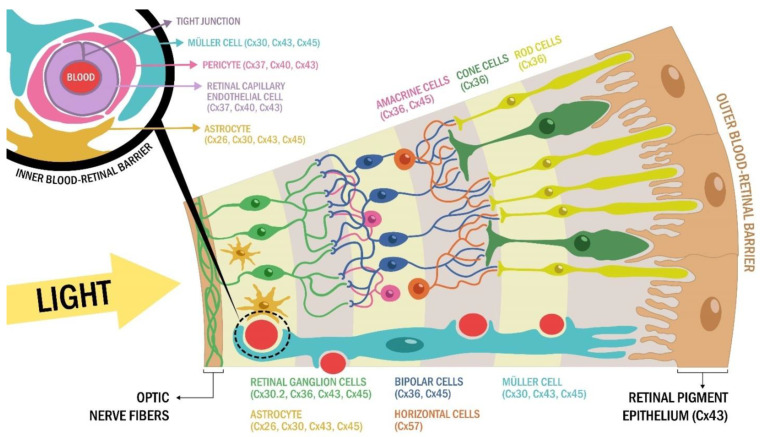
Connexin diversity in the adult retina. The retina is composed of neurons (cone and rod photoreceptors cells, horizontal cells, amacrine cells, bipolar cells, and retinal ganglion cells) and glia cells (Müller cells, microglia and astrocytes) arranged in different layers. Connected with each other, these cells form a complex circuit that converts light into electrical information in the brain. Briefly, light exposition hyperpolarizes photoreceptors (cone and rod cells) synapsing with bipolar cells. This electrical response is then propagated to retinal ganglion cells, whose axon projections makes up the optic nerve, transmitting the information to the brain. Because horizontal cells and amacrine cells mediate negative modulation of this circuit, the final retinal ganglion signal output corresponds to the net result of both excitatory (bipolar cell) and inhibitory signals (mainly amacrine cells). The main strategy for retinal signal transmission is electric transmissions, through gap junction intracellular communication (GJIC). These same GJIC are key players in the vascular function of the retina. In effect, retinal neurons coupled together with blood retinal barrier components (including retinal endothelial cells and retinal pigment epithelium) express diverse connexin proteins.

**Figure 2 ijms-22-03194-f002:**
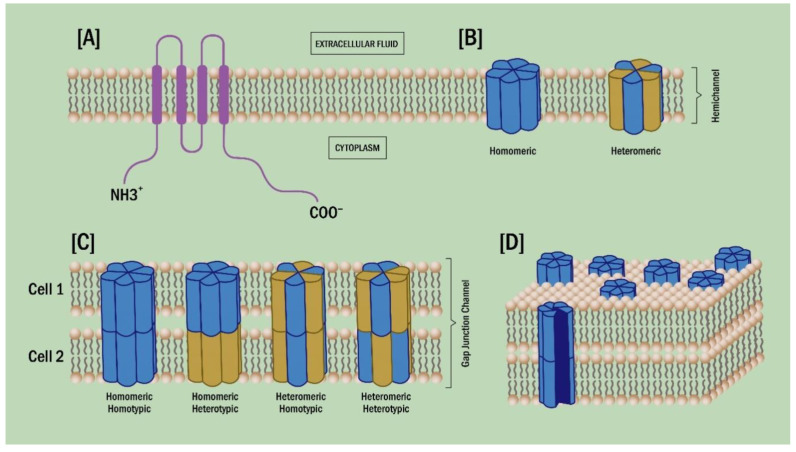
Connexins, hemichannels, and gap junction channels. (**A**) Connexins are membrane proteins with four transmembrane domains connected by two extracellular and one intracellular loops. Both amino- and carboxyl-termini of the protein are oriented towards the cytosolic domain of the cell. In mammals, between 19 and 21 different types of connexin genes have been reported. (**B**) The oligomerization of six connexins forms a hexamer named connexons or hemichannels (HCs). HCs can be homomeric or heteromeric if they are formed by the same or different connexin isoform, respectively. (**C**) HC interacting with opposing HC from neighboring cells can dock to form different types of gap junction channels (GJCs). (**D**) GJC can cluster to form a gap junction plaque.

**Figure 3 ijms-22-03194-f003:**
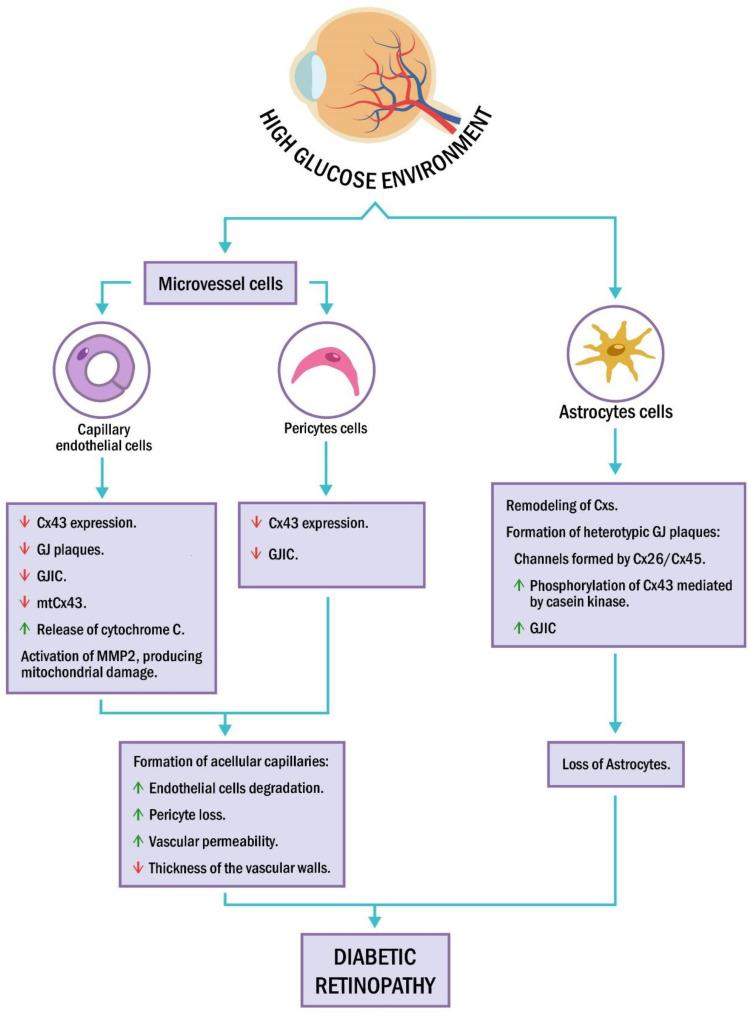
Connexin alterations reported in capillary endothelial cells, pericytes, and astrocytes during diabetic retinopathy (DR) in high glucose cellular environments.

**Table 1 ijms-22-03194-t001:** Connexin type expression in retinal cells.

Cell Type with Connexin and/or GJIC *	Connexin Type	Reference
Astrocyte	Cx43, Cx26, Cx30, Cx45	[54,55]
Müller cells	Cx43, Cx30, Cx30.3, Cx32, Cx43, Cx45, Cx46, Cx50	[56,57,58]
Endothelial cells	Cx30.2, Cx37, Cx40, Cx43	[59,60,61,62,63,64]
Pericytes	Cx37, Cx40, Cx43	[65,66,67]
Cone photoreceptors	Cx36	[68,69,70,71]
Rod photoreceptors	Cx36	[70,72,73]
Cone to rods	Cx36	[72,73,74,75]
Bipolar cells	Cx36, Cx45	[76,77,78]
Horizontal cell	Cx57	[79,80]
Bipolar to AII amacrine cell	Cx36, Cx45	[78,81,82]
Ganglion cell	Cx30.2, Cx36, Cx43, Cx45	[81,82,83,84]
AII Amacrine cell	Cx36, Cx45	[85,86,87,88]
AII amacrine cell to ganglion cell	Cx36	[89]
Retinal pigment epithelium	Cx43	[82,90]

* Gap Junction intercellular communication.

**Table 2 ijms-22-03194-t002:** Summary of Connexin modulator treatment strategies for diabetic retinopathy.

Connexin Modulator	Pharmacological Target of Connexin Modulator	Animal Model	Pathological Conditions	Effects	Reference
Peptide5	Cx43 mimetic peptide. Inhibits Cx43 HCs. Prevents HCs opening [182]	Adult albino Sprague-Dawley rats	Acute injury of retinal function model by induction of light damage.	Maintenance of photoreceptoral and postphotoreceptoral neurons function.	[183]
Peptide5	Cx43 mimetic peptide. Inhibits Cx43 HCs. Prevents HCs opening [182]	Human retinal pigment epithelial cells (ARPE-19)	High glucose and inflammatory conditions: exposition to IL-1β and TNFα.	Inhibition of cytokine release. Inhibition of the increase in ATP release.	[62,140]
Tonabersat	In low doses blocks Cx43 HCs without affecting GJC [184]. Also affect Cx26 GJs in Trigeminal Neurons [185].	Human retinal pigment epithelial cells (ARPE-19)	High glucose and inflammatory conditions: exposition to IL-1β and TNFα.	Inhibition of the release of cytokines IL-1β, VEGF, and IL-6 Inhibition of NLRP3 and cleaved caspase-1 complex formation.	[186]
Tonabersat	In low doses blocks Cx43 HCs without affecting GJC [184]. Also affect Cx26 GJs in Trigeminal Neurons [185].	Ex vivo human organotypic retinal culture	High glucose and inflammatory conditions: exposition to IL-1β and TNFα.	Inhibition of NLRP3 inflammasome complex assembly, Inhibition of Müller cell activation. Inhibition of release of IL-1β, IL-8 cytokines and VEGF.	[187]
Tonabersat	In low doses blocks Cx43 HCs without affecting GJC [184]. Also affect Cx26 GJs in Trigeminal Neurons [185].	Albino Sprague Dawley rats	Light-damaged retina model. Age-related macular degeneration in a spontaneous rat model of diabetic retinopathy.	Reduction of Inflammation Restauration of retinal electrical function.	[188]
Danegaptide (GAP-134, compound 9f)	Maintains GJIC under stressed conditions. Improves cell-cell communication. Improves gap junctional conductance [189]	Rat retinal endotelial cells	High glucose condition	Inhibition of apoptosis and excess vascular permeability.	[190]

## Data Availability

This review includes no original data.

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
