# Peer review of "An Update on Connexin Gap Junction and Hemichannels in Diabetic Retinopathy"

_ijms, 2021, doi:10.3390/ijms22063194_

Round 1
Reviewer 1 Report
(1) The review is written well and well organized. In the abstract initially, they say that "the expression of vascular connexin43 is decreased by prolonged hyperglycemia" and then the next statement says, "connexin channel blockers are promising tools for the development of pharmacological interventions for diabetic retinopathy". Even though these two statements are different it is kind of misleading in an initial read especially to be in the abstract as it is read by most people. (2) The authors can brief the functional role of glut receptors in their introduction part. (3) The authors can make a table of different treatment strategies currently in the trials (not clinical trials) for different cell types.Author Response
Please see the attachment.

Reviewer 2 Report
The thought behind this paper is very good. Although the style and presentation are good, the author(s) should reword sentences that have grammatical/punctuation errors. The authors have not given appropriate credit in citing the literature in very instance. The authors should endeavor to cite more original research in the revised version of the manuscript. This is a major concern about the manuscript that the authors should address.
Specific suggestions/comments raised by the reviewer:
Line 70: Did you mean to state "retinal microvascular endothelium or retinal capillary endothelium"?
Line 74: the author(s) should elaborate on what happens when these new pathological vessels rupture? Additionally, the author(s) should elaborate on the actual link between the new pathological vessels and retinal detachment.
Line 83: Render "the human," as "the human retina,"
Line 88: did you mean to say diabetes-invoked?
Line 96: Reword the sentence in order to render it comprehensible.
Lines 102 to 105: Similarity with lines 72 to 75. I consider it to be an unnecessary repetition.
Line 137: Bipolar cell is a retinal neuron, right?
Lines 149 – 204: The author(s) should endeavor to cite more original research in the revised version of the manuscript. The author(s) have not in every instance given proper credit in citing the literature. The author(s) tend to cite review articles by authors who did not contribute to a given discovery.
Lines 171 -172: Reword the sentence in order to render it comprehensible.
Lines 183 - 187: Reword these sentences in order to render them comprehensible.
Lines 188 - 190: Reword the sentence in order to render it comprehensible.
Lines 237: The author should cite the other studies that have shown that Cx43 suffers modifications under diabetic conditions affecting its interactions with structural proteins such as the tight-junction protein zonula occludens 1. The author(s) only cited one study.
Lines 254 - 258: Reword the sentence in order to render it comprehensible.
Line 279: C43 or Cx43?
Lines 350 - 355: Consider revising this run-on sentence.
Lines 381 - 385: Reword this sentence in order to render it comprehensible.
Line 391: "motive", the author(s) should use an alternate word for motive.
Reviewer 3 Report
The manuscript from González-Casanova et al. presents an interesting and well integrative perspective about the role of connexin channels, either hemichannels or gap junctions, in the pathophysiology of diabetic retinopathy. It describes, in an unusual but very attractive manner, the mechanisms and biological processes in which retinal connexins are involved. This review is grounded in very recent references which demonstrates the pertinence and timeliness of the topic and includes informative and attractive figures/diagrams. Overall, this reviewer thoroughly enjoyed reading this manuscript and found the information included informative, well organized and nicely presented, some minor issues should be addressed in order to improve the quality of the manuscript and make it more attractive for a wider audience.
Minor issues:
- As a suggestion: more information concerning the basis of the disease should be included for example: age of disease development, prevalence of diabetic retinopathy in Type 1 and Type 2, available therapeutic approaches.
- Table 1: “Cell type with GIJC” should be corrected to “Cell type with GJIC” or replaced to be broader and include more than gap junction intercellular communication and not exclude the possibility of positivity for Cx in other forms like hemichannels
- Please consider including a description of the genetic mice models of T1DM ( Akita and Akimba) and a possible explanation for the differences observed.
- Some typos were detected, please check lines: 140, 172, 203 and 383
Reviewer 4 Report
The review is focused on the role of the connexin gap junction in the development of diabetic retinopathy. However, the manuscript offers too general a description of the knowledge in this field and the lecture only provides didactic information, far from the aim of a review. For this main reason I advise against its publication in this journal.
This reviewer does not mean to be harsh and although authorized authors in the field of diabetes and gap junctions are adequately cited; the manuscript does not appear to provide a breakthrough or a clear point of view on this issue. More extensive documentation on the physiological role of the different connexins expressed in the different types of cells of the retina would have been necessary. On the other hand, the review focuses mainly on the Cx43 connexin, and although it is currently the most studied, the role of others should have been addressed in greater detail.
Reviewer 5 Report
In this manuscript, González-Casanova et.al., reviews an update on the role of Connexin Gap Junctions and Hemichannels in Diabetic Retinopathy (DR) and the subsequent emergence of potential new drugs. This review is a valuable addition to the field as it summarizes the existing literature on DR, connexin gap junctions, hemichannels, along with how DR affects vascular connexins and gap junction channels, connexins in retinal glial cells, and hemichannels in inflammation. The manuscript is well organized and figures are well presented. The article has interesting observations which are beneficial to researchers in the areas of DR and other retinal associated complications. However, the manuscript is quite confusing in parts, with syntax and grammar needing to be revised in several sections with a thorough proof check. Importantly, more appropriate original references need to be used in DR sections, along with further check on spellings and spacings throughout the manuscript. Some major and minor concerns discussed below.
Major concerns:-
-Line 91 - Describe role of AMPA receptors.
-Line 95 - Along with the mentioned retinal neurons, explain the effect of DR on ganglion cells, as they are stated several times in the review. Furthermore, explain the effect of DR on neurovascular unit, endothelial cells and pericytes in the introduction, as there are mentioned in the review later.
-Line 122 - Also add the role of advanced lipoxidation end-products (ALEs) and Müller cells in DR, with appropriate references.
-Line 129 - Update Table 1 caption to “Expression of different connexin types in retinal cells”. Clarify abbreviation of GIJC in Table 1. Also change “Endotelial” to “Endothelial” and “Retinal pigmented epithelium” to “Retinal pigment epithelium” in Table 1.
-Line 136 to 147 - Figure 1 nicely shows the expression of different connexins in retinal cells. This figure can be further enhanced by removing the grey background. Also, Figure 1 caption is confusing and needs to be rewritten with better syntax and English.
-Line 171 – Explain further on “regulation of these”
-Line 171 – 174 – Add further explanations and original references.
-Line 236 – Add more detail on “blockage of connexin hemichannels”
-Line 277 to 280 – Add further explanations and original references.
-Line 307 to 314 – Add appropriate and original references.
-Conclusion - Discuss further on role of connexins (neuronal, RPE etc.) and potential treatments which could advance DR treatment.
Minor concerns:-
-Line 66 - Add “it leads to” severe visual impairment.
-Line 85 - Clarify abbreviation of ERG
-Line 125-127 - Reword or separate into two sentences.
-Line 168 - Change “connexin” to “connexins”
-Line 183 - Reword with better syntax
-Line 189 - Change “activity is weak or practically zero” to “activity is minimized”
-Line 245 to 247 - Reword with better syntax
-Line 279 - Change “C43” to “Cx43”
-Line 316 - Specify the primary retinal endothelial cells.
-Line 371 - Change “Müller” to “Müller cells”.
Round 2
Reviewer 1 Report
The authors need to focus on their language, grammar, and spell check, following are few, the authors need to do a thorough revision of the written content to avoid these errors.
line 358 pharmecological- pharmacalogical
multiple lines -revesratrol- -Resveratrol
line 103 this transporter - GLUT1
Reviewer 4 Report
In this new version, the authors have considerably improved the quality of the manuscript. Could it be possible to add a section that refers to the role played by hemichannels in the neural cells of the retina in the context of diabetic retinopathy?
I think this information would be interesting because the work basically talks about the influence they have on astrocytes and vascular cells and does not mention other cell types where they have been shown to express themselves.
Reviewer 5 Report
González-Casanova et.al., have thoroughly addressed all the concerns raised in the initial manuscript and the efforts are greatly appreciated. The revised manuscript can deserve publication after making the minor changes mentioned below.
Minor changes:-
-Line 57-59, Recent review by Antonetti et.al., (Nat Rev Endocrinol (2021). https://doi.org/10.1038/s41574-020-00451-4) will be a good reference here.
-Line 95, Change “ganglionar cells” to “ganglion cells”.
-Line 111-113, Recent review by Augustine et.al., (Front. Endocrinol. 11:621938. doi: 10.3389/fendo.2020.621938) will be a good reference here.
-Line 114-115, Clarify the abbreviations of GJC and HC here.
-Line 119, change “conenxins” to “connexins”.
-Line 154, Change "retinal pigmented epithelium" to "retinal pigment epithelium"
-Line 227, Avoid the “( )” for the references.
-Line 266-268, Add an appropriate reference for OIR model.
-Line 314, Change “Annimal Model” to “Animal Model” in Table 2 headings.
-Line 358, Change “pharmecological” to “pharmacological”.
-Line 364, Replace “Last” with “Finally” or better word.
